# Pulsed Electric Field-Assisted Extraction of Inulin from Ecuadorian Cabuya (*Agave americana*)

**DOI:** 10.3390/molecules29143428

**Published:** 2024-07-22

**Authors:** Alejandra Rivera, Marcelo Pozo, Vanessa E. Sánchez-Moreno, Edwin Vera, Lorena I. Jaramillo

**Affiliations:** 1Departamento de Ingeniería Química, Facultad de Ingeniería Química y Agroindustria, Escuela Politécnica Nacional, Ladrón de Guevara E11-253, Quito 170525, Ecuador; alejandra.rivera@epn.edu.ec (A.R.); vanessa.sanchez@epn.edu.ec (V.E.S.-M.); 2Departamento de Automatización y Control Industrial, Facultad de Ingeniería Eléctrica y Electrónica, Escuela Politécnica Nacional, Ladrón de Guevara E11-253, Quito 170525, Ecuador; marcelo.pozo@epn.edu.ec; 3Departamento de Ciencia de los Alimentos y Biotecnología, Facultad de Ingeniería Química y Agroindustria, Escuela Politécnica Nacional, Ladrón de Guevara E11-253, Quito 170525, Ecuador; edwin.vera@epn.edu.ec

**Keywords:** inulin, cabuya (*Agave americana*), green extraction technique, optimization, pulsed electric field, FTIR, TGA

## Abstract

Inulin is a carbohydrate that belongs to fructans; due to its health benefits, it is widely used in the food and pharmaceutical industries. In this research, cabuya (*Agave americana*) was employed to obtain inulin by pulsed electric field-assisted extraction (PEFAE) and FTIR analysis confirmed its presence. The influence of PEFAE operating parameters, namely, electric field strength (1, 3 and 5 kV/cm), pulse duration (0.1, 0.2 and 0.5 ms), number of pulses (10,000, 20,000 and 40,000) and work cycle (20, 50 and 80%) on the permeabilization index and energy expenditure were tested. Also, once the operating conditions for PEFAE were set, the temperature for conventional extraction (CE) and PEFAE were defined by comparing extraction kinetics. The cabuya meristem slices were exposed to PEFAE to obtain extracts that were quantified, purified and concentrated. The inulin was isolated by fractional precipitation with ethanol to be characterized. The highest permeabilization index and the lowest energy consumption were reached at 5 kV/cm, 0.5 ms, 10,000 pulses and 20%. The same extraction yield and approximately the same amount of inulin were obtained by PEFAE at 60 °C compared to CE at 80 °C. Despite, the lower amount of inulin obtained by PEFAE in comparison to CE, its quality was better because it is mainly constituted of inulin of high average polymerization degree with more than 38 fructose units. In addition, TGA analyses showed that inulin obtained by PEFAE has a lower thermal degradation rate than the obtained by CE and to the standard.

## 1. Introduction

Inulin is a carbohydrate that belongs to the subgroup of fructans. It is composed of terminal glucose and fructose molecules linked by β (2 → 1) bonds [1,2]. Inulin is used as a pharmaceutical excipient, prebiotic and substitute for sugar and fat in dairy and meat products, respectively [3]. Inulin consumption reduces the incidence of diseases such as colon cancer, obesity and diabetes; the last of these was the seventh greatest cause of death, claiming approximately 4.0 million people, in 2017 [4,5,6]. Other benefits related to inulin consumption are the improvement in calcium absorption, the relief of constipation and the decrease in cholesterol and triglyceride levels in the blood [5,7].

The industrial production of inulin is carried out by conventional extraction by means of hot water, within which inulin diffuses out of the cell [8]. Leaching at high temperatures increases the solubility of inulin and denatures the proteins that are in the cell membrane, thereby improving the transport of inulin [8,9]. This extraction requires long times (1.5–2.0 h) and high processing temperatures (70–80 °C) [10] that can be reduced using non-conventional techniques [11]. Currently, non-conventional techniques for non-thermal food processing such as ultrasound, microwave, ionizing radiation, pulsed magnetic field and pulsed electric field (PEF) have widely been studied, since they can improve conventional processing and food quality [10,12].

Chicory roots (*Cichorium intybus*) and Jerusalem artichoke (*Helianthus tuberosus*) are mainly employed for the industrial production of inulin [7,10]; however, other species like dahlia (*Dahlia* spp.), burdock (*Arctium lappa*), jicama (*Smallanthus sonchifolius*) and cabuya (*Agave americana*) have been studied for inulin extraction [2,13,14]. Cabuya is an Andean-native plant that belongs to the *Agavaceae* family, traditionally used as a medicinal plant [2]. Despite many studies reporting the extraction of inulin or fructans from agave species [15,16,17], only two reported this extraction from *Agave Americana* using CE [2,18], and there is no information about PEF-assisted extraction (PEFAE) from cabuya.

In recent years, PEF, as a non-thermal pretreatment, has been used for many applications, such as pasteurizing liquids, inactivating enzymes, wastewater treatment and to improve different processes of industrial value such as pressing, drying and extraction. [19]. 

In PEFAE, short-duration pulses (in the order of ms) with high electric field strengths (0.1–10 kV/cm) are applied to food placed between two electrodes [20]. The mechanism of PEFAE is explained according to the electromechanical model: when the cells are exposed to an electric field (E), the reorganization of charges in the cell and through the membrane is induced as observed in Figure 1, increasing the transmembrane potential. The charges compress the membrane that is electrically non-conductive because of the attractive forces between charges of opposite signs, resulting in a deformation of the membrane. If the transmembrane potential exceeds a critical value, the charges on both sides of the membrane produce electrocompressive forces that can cause dielectric rupture of the membrane [21], creating new pores [12]. This rupture is named electroporation, which is the most accepted mechanism of electroplasmolysis of cells [22]. Electroplasmolysis increases electrical conductivity, diffusion, heat and mass transfer coefficients; compressibility of plant tissues; and cell membrane permeability, thereby increasing the extraction efficiency of cellular compounds [22]. The extraction speed in PEFAE depends on operating parameters such as field strength, number of pulses, pulse duration and duty cycle, and external conditions such as agitation, temperature, solid/liquid ratio, size-related effects, etc. [23].

PEFAE of inulin has been studied in chicory roots, which have an inulin content of between 13 and 23% [10,24]. In the cited studies, it was concluded that the permeabilization of the tissue and the extraction of soluble matter are improved when PEF is applied. In the study carried out at laboratory scale by Loginova et al. [24], the use of PEF improved inulin extraction at temperatures between 20 and 40 °C, and acceleration of soluble matter extraction concerning conventional extraction (CE) was observed. Zhu et al. [10], in their pilot-scale research, also found that by applying PEF and extracting inulin at 60 °C, the purity of the extract and the concentration of inulin are comparable to those obtained by CE at temperatures between 70 and 80 °C. Based on these studies, it is considered that using PEFAE of inulin from cabuya is a viable alternative.

The purpose of this research was to determine the best pretreatment conditions with a monopolar pulsed electric field in the extraction of inulin from Ecuadorian cabuya. The effect of the electric field strength, number of pulses, pulse duration, duty cycle and extraction temperature were determined. Likewise, the characteristics of the extract and inulin obtained with conventional and non-conventional extraction were compared.

## 2. Results and Discussions

### 2.1. Determination of the Electric Field Strength, Number of Pulses, Pulse Duration and Duty Cycle

#### 2.1.1. Determination of the Characteristic Frequency of Cabuya 

Figure 2 shows the frequency scan of the system (cabuya between the plaques). It is observed that at low frequencies, the impedance of the damaged tissue is lower than the impedance of the intact tissue. This is due to the rupture of cell membranes, which under normal conditions (intact tissue) act as capacitors that prevent the flow of electric current through the intracellular medium [25]. Thus, the current flows only through the extracellular medium and the conductivity is related only to the ion concentration in this medium. 

On the other hand, in the high-frequency range, the impedances of intact and damaged tissues are similar. This is because when frequency increases, cell membranes become less resistant to the flow of current in the intracellular fluid [25], so they become conductors of the electric current. Thus, at very high-frequency values, the membranes are completely short-circuited, and the conductivity represents the contribution of the extracellular and intracellular medium [25,26].

Figure 2 shows that the impedance modules of damaged tissues do not depend on frequency. Additionally, it is observed that the angles of impedance in the whole frequency range have negative values, which means that the system behaves in a capacitive manner. This result was expected since the measurement system used simulates a capacitor in which two conductive bodies (plates) are separated by a dielectric (cabuya + air) [27].

The impedance curves of intact tissues in Figure 2 show a change in slope and concavity at 2 kHz. At this frequency, there is also a difference between the impedances of intact and damaged tissues for the two meristems. This difference decreases as the frequency increases, which makes cell membrane damage imperceptible. Also, it is observed that at 2 kHz, there is a convergence change in the phase curves of the intact tissues. Therefore, the characteristic frequency of the system for both meristems used in Section 2.1.2 (Meristem 1) and Section 2.1.3 (Meristem 2), is approximately 2 kHz. This frequency falls between 1–5 kHz, this is the range used by Vorobiev and Lebovka [28] to measure the electrical conductivity of vegetable materials, because within this range, the permeabilization of the tissue induced by the PEF is detectable.

#### 2.1.2. Selection of Electric Field Strength, Number of Pulses and Pulse Duration

The standardized Pareto diagram for the permeabilization index is shown in Figure 3a. In this figure, it is observed that electric field strength has a significant effect on the permeabilization of the cell membrane. The result is consistent with that obtained by Loginova et al. [24], who determined that the degree of permeabilization increased after pretreatment with PEF and the effect was more pronounced at higher electric field intensities. 

Figure 3b shows that at 0.5 ms, the best electric field intensity was 5 kV/cm. This value is within the range of 0.1 and 5 kV/cm when the plant cells are electroporated, according to Vorobiev and Lebovka [28]. It is important to mention that this parameter depends on the types of cells because it varies with the cell density, size, position and arrangement of the cells [29]. Furthermore, in Figure 3b, it is observed that at 0.5 ms the permeabilization index of the cell membrane increases with the electric field strength. This is because the electric field strength is proportional to the accumulated charge in the membrane, as a capacitor. As the electric charge increases, the transmembrane potential increases and when it exceeds the critical value, electroporation occurs. When applying the electric field strength higher than necessary to reach the critical value, the permeabilized area in the cell membrane increases [29].

Figure 3a shows that the pulse duration has a significant effect on the permeabilization index; this was also found by De Vito et al. [19]. This is because, to exceed the critical value, the pulse duration must be longer than the charging time of the membrane. The effect of the pulse duration depends on the intensity of the electric field. This was confirmed since the Pareto diagram of standardized effects of the permeabilization index (Figure 3a) shows a significant effect of the interaction of these variables at a constant number of pulses [11]. Also, in Figure 3a, it is observed that the permeabilization index is not influenced by the number of pulses. This is consistent with findings by Dörnenburg and Knorr [30], who determined that the effect of electric field strength is more significant than the number of pulses, in the release of anthraquinone and amaranthine from *Morinda citrifolia* and *Chenopodium rubrum* cells, respectively. In addition, the results obtained by Lebovka et al. [31] suggest that using a small number of pulses and high electric field intensity has the most significant permeabilization effects in apple cells.

The standardized Pareto diagram for energy expenditure is presented in Figure 3c. It suggests a significant and direct effect of the electric field intensity, pulse duration and their interaction on the energy expenditure. However, for this response variable, the number of pulses and their interaction with the other variables have a significant effect. This result was expected since, according to Gachovska et al. [32], it is desirable to have the highest amount of energy per pulse and fewer pulses to minimize energy expenditure.

Although the number of pulses is not a significant variable for tissue permeabilization (Figure 3a), it influences energy expenditure (Figure 3c). Thus, by reducing the number of pulses from 40,000 to 10,000, the energy spent per kilogram of treated meristem decreases from 1210.33 to 302.88 kJ/kg. Also, at 10,000, the permeabilization index is higher (0.57) than at 40,000 pulses (0.43). So, 10,000 pulses allow for greater permeabilization of the cell membrane.

The energy used for the permeabilization of cabuya is 302.88 kJ/kg. This value is greater than the energy used in the permeabilization of carrots (9 kJ/kg) and sugar beets (5–6 kJ/kg) [33]. The energy used is in the range of 40–1000 kJ/kg required for the inactivation of bacteria [28].

The energy expenditure, in this study, is higher because different PEF equipment was used. The electrodes were in contact with the plant tissue in the studies cited, while in the present study, the electrodes were not in contact with the cabuya slices. Therefore, the dielectric and capacitance of the capacitor formed in the cited studies (plant tissue) are different from those used in the present study (plant tissue + air). The capacitance is greater without the presence of air between the electrodes; therefore, the voltage and the energy applied to reach the same electric charge are less [27].

Based on the statistical analyses performed, it was determined that the best pretreatment conditions were 5 kV/cm of electric field intensity, 0.5 ms of pulse duration and 10,000 pulses corresponding to 5 s of effective treatment at 50% of duty cycle. The treatment time applied was greater than 0.3 s, which was used for inulin extraction from chicory roots in the study of Loginova et al. [24].

#### 2.1.3. Selection of the Duty Cycle 

Another variable that influences the pretreatment with PEF is the duty cycle, which is the percentage of pulse duration for the period (total time between on and off) [27]. The mean diagrams of the duty cycle as a function of the permeabilization index (a) and energy expenditure (b) are shown in Figure 4. The figure shows that the cellular integrity of the treated tissue is reduced when the lowest value of the duty cycle is used, while energy expenditure increases as the duty cycle increases. Each value of the duty cycle corresponds to only one frequency and one-time interval between pulses: for 20, 50 and 80%, the frequencies and intervals of time are 400 Hz and 2 ms, 1000 Hz and 0.5 ms and 1667 Hz and 0.12 ms, respectively. Therefore, the best duty cycle found was 20%, corresponding to the lowest frequency and the highest time interval at an effective treatment time of 5 s. When considering the time interval between pulses, the results are consistent with those obtained by Lebovka et al. [34], who determined that a time interval between pulses of 60 s accelerates the permeabilization of apple tissue compared to 10^−2^ s for the same effective treatment time. 

Considering the frequency, the results correspond to those obtained by Asavasanti et al. [35], who suggested that lower frequencies (<1 Hz) result in greater permeabilization of onion tissue than higher frequencies. At low frequencies, the time interval between pulses is enough for diffusion and cyclic processes to occur. Through these processes, the cytoplasmic fluid (high conductivity) moves into the extracellular space (low conductivity) through the pores formed by the PEF [35]. By increasing the conductivity of the extracellular fluid around the broken cells, it becomes the best route for the next electrical pulse to travel and reach more intact cells [35].

At low frequencies, the speed of cyclosis decreases gradually, thereby accelerating physical damage and improving the transport of cytoplasmic fluid to the extracellular fluid. In addition, cyclosis helps to supply additional ions to the edge of the cell wall near the site of membrane rupture, which helps in maintaining a concentration gradient that benefits ion flow [35]. At high frequencies, the speed of the cyclosis is reduced to zero almost instantaneously and the periods are too short for the redistribution of the ions to change the conductive path to adjacent intact cells [35].

### 2.2. Study of the Temperature Effect on Pulsed Electric Field-Assisted Extraction and Conventional Extraction 

The extraction kinetics presented as a normalized concentration of solute with respect to the extraction time for the CE (a) and the PEFAE (b) at different temperatures are shown in Figure 5. The extraction kinetics shows that the highest concentration of soluble matter is achieved at 80 °C for CE (Figure 5a) and at 60 °C for PEFAE (Figure 5b). The positive effect of PEFAE for CE is observed at temperatures below 60 °C, while at 80 °C, the effect of pretreatment is inhibited [24], as observed in Figure 5c.

Figure 6 shows that the effective diffusion coefficient values for PEFAE are higher than those for CE in the range of 30 to 60 °C. On the other hand, at 70 °C, the effective diffusion coefficient was almost the same for slices with (2.02 × 10^−10^ m^2^/s) and without (1.98 × 10^−10^ m^2^/s) pretreatment.

From the analysis performed, it was defined that the best temperatures in CE and PEFAE were 80 and 60 °C, respectively. At these temperatures, the same concentration of normalized solute B was reached at 2, as seen in Figure 5c. By using a lower temperature for PEFAE, lower energy consumption is achieved.

The effective solute diffusion coefficient values shown in Table 1 are within the diffusion range of a solute contained in a solid matrix to a solvent (10^−9^–10^−10^) [36]. An increase in the coefficient is observed due to electroporation at 60 °C. Table 1 shows that the lowest concentration of free fructose was obtained at 80 °C by CE because the integrity of the inulin chains was maintained. Also, the concentration of free fructose at 60 °C is higher for CE than for PEFAE. The concentration of inulin at 60 °C by PEFAE was higher than the obtained at 60 °C by CE, and it was similar to that obtained at 80 °C by CE.

### 2.3. Comparison of the Inulin Powder Obtained by Pulsed Electric Field-Assisted Extraction and Conventional Extraction

#### 2.3.1. Comparison of Inulin Recovery and Production

The recovery and production of inulin are shown in Table 1; as is seen, the inulin obtained by PEFAE has lower values of inulin recovery and production than those obtained by CE at 60 and 80 °C. The inulin recovery values obtained in this study are higher than those obtained from *Dahlia* L. spp. (3.22%) [37]. On the other hand, the inulin production values obtained in this study are lower than those obtained from *Jerusalem artichoke* (12%) [38] and *Helianthus tuberosus* L. [39].

The inulin obtained was fractionated in inulin of high average polymerization degree (${In}_{H-APD}$) and low average polymerization degree (${In}_{L-APD}$). As shown in Table 1, more than half of the inulin obtained by means of PEFAE precipitated at 20 and 40% *v*/*v* ethanol. According to Escobar-Ledesma et al. [2], it has a degree of polymerization between 38 and 59. Inulin obtained by PEFAE has a greater quantity of ${In}_{H-APD}$ than CE; this kind of inulin has a significant effect on calcium absorption and flora improvement in the most distal parts of the colon because it is fermented twice. In addition, it can be used as a fat substitute because of its ability to form gels [40].

The difference between the recovery and production values obtained by using PEFAE and CE at 60 °C can be attributed to the different size and molecular structure of the inulin extracted, since these features influence the precipitation of polysaccharides in ethanol [41]. Additionally, the recovery and production of inulin could have been affected by the presence of proteins that also precipitate with ethanol [42]. Inulin powder had proteins, as observed in the FTIR analysis. This contamination is lower for inulin obtained by PEFAE since PEF denatures proteins [43].

#### 2.3.2. Comparison of Inulin Spectra and Thermograms

The infrared spectra of the isolated inulin powders from cabuya (${In}_{H-APD}$ and ${In}_{L-APD}$) are shown in Figure 7. In both spectra, a broad band is observed between 3550 and 3230 cm^−1^, which is related to the stretching of the hydroxyl group in the fructose. The following bands, in the ranges of 2940–2915 cm^−1^ and 2870–2840 cm^−1^, are attributed to CH_2_ stretching. The stretching of the CH bonds was the reason for the peaks between 2900 and 2800 cm^−1^ [44].

In the spectrum of ${In}_{H-APD}$ obtained by CE at 60 °C in Figure 7a, the band at 2340 cm^−1^ is attributed to a false signal provided by the CO_2_ in the atmosphere [44]. Additionally, it is observed in the spectra of ${In}_{H-APD}$ and ${In}_{H-APD}$ bands at 624 cm^−1^ and 1615 cm^−1^, respectively, which were related to primary amines in the protein deformation phase [39].

In both spectra of Figure 7, the bands between 1475–1445 cm^−1^ and 1413 cm^−1^ are related to the bending of CH_3_ bonds and to the deformation of the CH_2_OH bond found in the fructose ring, respectively [45]. The bands between 1270–1060 cm^−1^ and 1200–750 cm^−1^ are associated with the stretching of the COC bond [45] and the CO bonds of alcohols [39], respectively. The bands around 930 cm^−1^ are related to the α-D glucopyranose residue in the carbohydrate chain [39] and those observed between 895–892 cm^−1^ and 874 cm^−1^ are related to 2-ketofuranose ring vibration [45]. The movements of 2-ketose and hydrogen bonds of the alcohols were the reason for the bands around 817 and 650 cm^−1^, respectively. Furthermore, the bands below 800 cm^−1^ are related to the movements of the hydrogen bonds in water [45]. Similar bands of standard inulin and the substances extracted confirm the presence of inulin.

Figure 8 shows the TGA thermograms of the obtained inulins (${In}_{H-APD}$ and ${In}_{L-APD}$) and the standard with several mass loss events. In both thermograms, three loss stages can be identified. The first stage occurs at temperatures below 130 °C and is due to the loss of volatile components and moisture content of the sample [2]. The second stage occurs between 200 and 250 °C; this represents the thermal degradation of branches of the inulin molecules [46]. The third stage occurs up to 290 °C and is associated with the thermal degradation of the main chains of the inulin molecules [47].

As seen in Table 2, all inulin obtained from cabuya has lower degradation percentages than standard inulin obtained from chicory. Also, inulin obtained by PEFAE has lower percentages than that obtained by using CE.

The thermal degradation rate of the inulin obtained decreases slowly against the standard. This decrease begins at 200 °C and is associated with the presence of branched chains of fructans in agave species [46]. This decrease is observed in the inulin obtained with PEFAE, as seen in Figure 8. The main reason is because PEF changes the intensity of the intermolecular forces, which influences the movement, the structural order of the chains, and therefore the thermal stability of inulin [48].

## 3. Materials and Methods

### 3.1. Materials and Reagents

The meristems of cabuya were harvested in San Antonio de Pichincha, Quito—Ecuador. They were between 10 and 12 years old and 1.20 to 2.00 m in height. The meristems were cut into slices of 2 mm in thickness and frozen for a week.

The spectrophotometric method proposed by Saengkanuk et al. [49] was used to determine the concentration of inulin in the extract with the following reagents: sodium meta periodate (Merck, Darmstadt, Germany, purity ≥ 99.0%), citric acid monohydrate (Merck, purity ≥ 99.5%), hydrated trisodium citrate (Merck, purity ≥ 99.5%) and potassium iodide (Merck, purity ≥ 99.0%). Sulfuric acid (J.T. Baker, Phillipsburg, NJ, USA, purity ≥ 97.99%) and sodium carbonate (BDH, Louisville, TN, USA, purity ≥ 98.0%) were used for the hydrolysis of inulin. The D-fructose standard (BDH, purity ≥ 98.9%) was food grade.

The method proposed by Li et al. [50] was used to purify the extract with the following reagents: calcium hydroxide (Baker Analyzed, purity = 97.6%), orthophosphoric acid (PROLABO, Tokyo, Japan, purity = 97.6%) and peroxide hydrogen (PROLABO, purity = 30.0%). Commercial chicory inulin (Beneo Orafti^®^ GR, Zeitz, Germany, with the degree of polymerization between 10 and 23) was used as standard.

### 3.2. Determination of Electric Field Strength, Number of Pulses, Pulse Duration and Duty Cycle

To determine the best values of the electric field intensity, number of pulses, pulse duration and duty cycle, the permeabilization index or Factor Z and the energy expenditure (relationship between the energy and normalized soluble matter content) were used as response variables. The permeabilization index was calculated with Equation (1) and the energy expenditure with Equation (4). The permeabilization index was calculated by means of the electrical conductivities and the electric resistances. The electric resistances were obtained by measuring the module and phase with LCR equipment (GW Instek, New Taipei City, Taiwan, model LCR-821) at the characteristic frequency of the Cabuya–Plates system. To do this, a pair of square bakelite electrodes, 15 mm per side, were attached to the LCR meter. Between the bakelite electrodes, cabuya slices approximately 1.5 cm in size and 2 mm thick were placed.
(1)$$Z=\frac{\sigma -\sigma_{i}}{\sigma_{d}-\sigma_{i}},$$
where Z is the permeabilization index, σ is the electrical conductivity of the tissue pretreated by PEF (S/cm), *σ*_i_ is the electrical conductivity of the intact tissue (S/cm) and *σ*_d_ is the electrical conductivity of the damaged tissue (S/cm). The damaged tissue was obtained after freezing the intact tissue at −10 °C for 72 h and thawing it at room temperature for 2 h [51]. The electrical conductivities were determined with Equation (2) [52].
(2)$$\sigma =\frac{d}{A\times R},$$
where d is the thickness of the slice (cm), A is the contact area of the tissue with the electrodes (cm^2^) and R is the electrical resistance (Ω) that was calculated with Equation (3) [27].
(3)R=I×cosθ,
where I is the impedance module (Ω) and θ is the angle of the impedance (rad) that was obtained with the LCR meter at the characteristic frequency of the cabuya. Energy expenditure was calculated with Equation (4) [52].
(4)$$W=\frac{U_{rms}\times I_{rms}\times N\times \tau }{B},$$
where W represents the energy expenditure (J); U is the effective value of the voltage (V); *I* is the effective value of the current intensity (A) that was measured with a digital oscilloscope (Tektronix, Beaverton, OR, USA, TDS2022C) with a current tip AC/DC, Fluke, 80i-110S; N is the number of pulses; τ is the duration of the pulse (s); and B is the content of normalized soluble matter, factor B- or °Brix-normalized.

#### 3.2.1. Determination of the Characteristic Frequency of the Cabuya–Plates System

The characteristic frequency was determined by plotting a Bode diagram, the impedance module and phase of the Cabuya–Plates system (also called transfer function) as a function of the frequency, as seen in Figure 9. These values are the characteristic responses of the different elements and compounds of nature when they are subjected to electrical signals of different amplitudes and frequencies.

The impedance module and phase were obtained with the LCR meter, at a sinusoidal voltage of 1 V in the range of 20–200,000 Hz. For the measurement, cabuya slices of two different meristems were used for the experimental tests of Section 3.2.2 (Meristem 1) and Section 3.2.3 (Meristem 2).

#### 3.2.2. Selection of Electric Field Strength, Number of Pulses and Pulse Duration

Pretreatment conditions were studied by applying an electric field of rectangular monopolar pulses to the cabuya slices (Figure 9). The electric field was generated by pulsed electric field equipment (DACI-EPN, Quito, Ecuador, Flyback Prototype) that provides pulses of 1 to 5 kV for 1 to 60 s, frequency of 1 to 5 kHz, 20 to 80% duty cycle and 1 cm of distance between electrodes.

A 3^3^ factorial experimental design was employed: three electric field strengths (1, 3 and 5 kV/cm), three pulse durations (0.1, 0.2, 0.5 ms) and three numbers of pulses (10,000, 20,000 and 40,000) were tested. The response variables were the permeabilization index and the energy expenditure; response surface graphs were obtained to determine the best conditions of the three studied variables. The duty cycle was set at 50%. The permeabilization index was calculated with Equation (1) [28].

The energy spent per kilogram of treated slices was calculated with Equation (5) [52].
(5)$$E=\frac{U_{rms}\times I_{rms}\times N\times \tau }{m\times 1000},$$
where E represents the energy spent per kilogram of treated slices (kJ/kg); m is the mass of slices pretreated with PEF (kg).

The effective value of the voltage for monopolar pulses is calculated employing Equation (6) [27].
(6)$$U_{rms}=v\times \surd \partial ,$$
where V is the applied voltage (V) and δ is the duty cycle (see Section 3.2.3).

Factor B was calculated with Equation (7) [24].
(7)$$B=\frac{{}^{\circ}Brix-{}^{\circ}{Brix}_{i}}{{}^{\circ}{Brix}_{f}-{}^{\circ}{Brix}_{i}},$$
where °Brix is the content of soluble matter in the extract at 60 min, °Brix_i_ is the content of initial soluble matter in the extract measured immediately after placing the slices of cabuya in water, and °Brix_f_ is the final soluble matter content in the extract that was determined after 2 h of extraction at 80 °C. A digital refractometer (Optic Ivymen System, Comecta, Barcelona, Spain, ABBE WYA-2S) was used to measure the brix degrees.

The extract was obtained from a diffusion process of 30 g of cabuya slices (54 slices) in 90 mL of distilled water at 30 °C with constant magnetic stirring at 240× rpm, for 60 min, in 250 mL precipitation vessels [24].

#### 3.2.3. Duty Cycle Selection

The duty cycle influences energy expenditure during pretreatment; therefore, it was necessary to select the duty cycle that allows for obtaining the best permeabilization index and energy expenditure. A completely random design with three levels of duty cycle (20, 50 and 80%) and three repetitions was applied. The effect of different duty cycles was compared by analysis of variance (ANOVA) using the Tukey HSD method with 95% confidence. The levels correspond to the working limits of the equipment used to generate the pulsed electric field. Each duty cycle value corresponds to only one frequency and time interval between pulses. The optimal conditions found by the procedure described in Section 3.2.2 were set. The statistical analyses studied in this research were carried out with the software STATGRAPHICS Centurion XVI.I, version 16.1.03.

### 3.3. Study of the Temperature Effect on Extraction Assisted by Pulsed Electric Field and Conventional Extraction

The effect of temperature on conventional extraction (CE) and extraction assisted by pulsed electric field (PEFAE) at the best pretreatment conditions found in Section 3.2 was evaluated by plotting the kinetics of inulin extraction from cabuya (factor B vs. time). The temperatures evaluated were 30, 40, 50, 60, 70 and 80 °C [24] and the best temperature was the lowest at which the maximum inulin extraction was reached, for both extraction types (Factor B equal to 1) after 2 h of extraction.

Extracts were obtained by PEFAE and CE at the best temperature. In addition, an extract was obtained by CE at the best temperature of the PEFAE to evaluate both extraction techniques at the same temperature. The CE was performed at the same conditions as PEFAE but without pretreatment. For each extraction, the effective diffusion coefficient and the content of fructans were determined.

The effective diffusion coefficient was obtained by adjusting the experimental factor B values to the model used by Loginova et al. [24]. The content of soluble matter in the extract was measured at 0, 1, 2, 4, 6, 9, 12, 15, 30, 60, 90 and 120 min [2]. Extractions were performed with the solid/liquid ratio and the stirring speed used in Section 3.2.

The concentration of inulin in the extract was determined with the spectrophotometric method presented by Saengkanuk et al. [49]. In this method, the concentration of inulin was obtained by subtracting the concentration of free fructose from the total fructose concentration and multiplying by a correction factor equal to 0.995. The absorbance measurements were performed with a UV/VIS spectrophotometer (Merck, Darmstadt, Germany, Spectroquant^®^Prove 100) at a 350 nm wavelength.

### 3.4. Comparison of the Inulin Powder Obtained through Extraction Assisted by Pulsed Electric Field and Conventional Extraction

The percentages of recovery and production were calculated to compare the inulin powder obtained by PEFAE and CE. Also, the spectra and thermograms of the obtained inulin and the standard inulin were obtained. This comparison allows for verifying whether the pretreatment with PEF might be an alternative CE in the extraction of compounds [12].

#### 3.4.1. Purification of Extracts

To purify the extracts, the proteins were precipitated at pH 11 with calcium hydroxide and then at pH 8 with a solution of orthophosphoric acid 1 M to remove excess calcium hydroxide [50]. The extracts were refrigerated at 4 °C for 12 h and centrifuged at 7100 rcf for 30 min to eliminate impurities [53]. Then, to bleach the solution, hydrogen peroxide solution of 0.9% *v*/*v* was added [50]. Subsequently, the extracts were concentrated to 30 °Brix on a heating plate at 60 °C and cooled to room temperature [54].

#### 3.4.2. Obtaining Inulin Powder

Four fractions of inulin were obtained from the extracts by ethanol precipitation at 20, 40, 60 and 80% *v*/*v*. For this, absolute ethanol was added to the extract until it reached a concentration of 20% *v*/*v* [2] and was stored at 2 °C for 24 h to facilitate precipitation. Then, the solution was centrifuged at 3000 rcf for 15 min and the inulin obtained was dried at 50 °C for 24 h [54]. The other inulin fractions were obtained by increasing the concentration of ethanol in the supernatant to 40, 60 and 80% *v*/*v*. The solid obtained at 20, 40 and 60% *v*/*v* of ethanol corresponds to the inulin of high average polymerization degree (${In}_{H-APD}$) and the solid obtained at 80% *v*/*v* ethanol corresponds to inulin of low average polymerization degree (${In}_{L-APD}$).

#### 3.4.3. Comparison of the Recovery, Production, Thermograms and Spectra of the Inulin Obtained through Pulsed Electric Field-Assisted and Conventional Extraction

Inulin recovery ($\eta_{recovery}$) and production yield ($\eta_{production}$) were calculated with Equations (8) and (9), respectively.
(8)$$\eta_{recovery}=\frac{m_{\mathrm{powder}\mathrm{of}\mathrm{inulin}}}{m_{\mathrm{inulin}\mathrm{in}\mathrm{the}\mathrm{extract}}}\times 100,$$
(9)$$\eta_{production}=\frac{m_{\mathrm{powder}\mathrm{of}\mathrm{inulin}}}{m_{\mathrm{wet}\mathrm{slices}\mathrm{of}\mathrm{cabuya}}}\times 100,$$
where $m_{\mathrm{powder}\mathrm{of}\mathrm{inulin}}$ is the amount of inulin recovered, $m_{\mathrm{inulin}\mathrm{in}\mathrm{the}\mathrm{extract}}$ is the amount of inulin in the extract and $m_{\mathrm{wet}\mathrm{slices}\mathrm{of}\mathrm{cabuya}}$ is the amount of cabuya slices used for extraction [38]. The recovery and production of the inulin obtained with CE were compared with those obtained by PEFAE.

Additionally, inulin powders were analyzed by Fourier-transform infrared spectrometry (FTIR) using an FTIR spectrometer (Perkin-Elmer, Waltham, MA, USA, Spectrum one) and thermogravimetry (TGA) with a thermal analyzer (Shimadzu, Kyoto, Japan, TGA-50). The spectra and thermograms of the inulin obtained by CE, PEFAE and standard inulin were compared.

## 4. Conclusions

Inulin from cabuya was obtained by PEFAE with an electric field strength of 5 kV/cm, 10,000 pulses, 0.5 ms of pulse duration and 20% of duty cycle. The use of PEF as pretreatment reduced the extraction temperature (from 80 °C to 60 °C) and energy consumption. The effective solute diffusion coefficient for PEFAE (2.36 × 10^−10^ m^2^/s) was higher than the diffusion coefficient for CE at 60 °C (1.68 × 10^−10^ m^2^/s). FTIR analyses confirmed the presence of inulin in the isolated substances obtained by PEFAE and CE. TGA analyses showed that inulin obtained by PEFAE has a lower thermal degradation rate than inulin obtained by CE and the standard. Even though the amount of inulin (recovery and production values) obtained by PEFAE was lower than that obtained by CE, its quality was better because it mainly constituted inulin of high average polymerization degree. This inulin is made up of more than 50% of inulin molecules that have more than 38 fructose units.

## Figures and Tables

**Figure 1 molecules-29-03428-f001:**
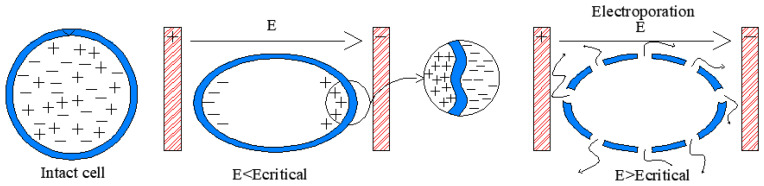
Scheme of cell permeabilization.

**Figure 2 molecules-29-03428-f002:**
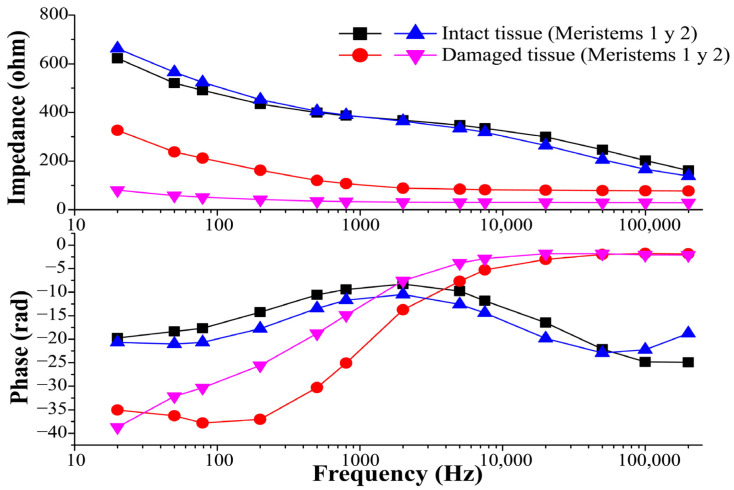
Impedance and phase of the system (Cabuya–Plaques).

**Figure 3 molecules-29-03428-f003:**
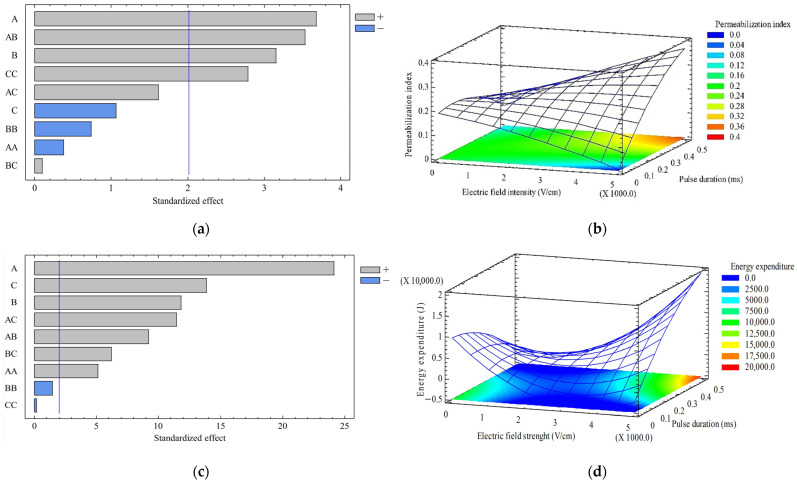
Pareto diagrams of standardized effect of electric field intensity (A); pulse duration (B); number of pulses (C); permeabilization index (**a**) and energy expenditure (**c**); and surfaces of response corresponding to the permeabilization index (**b**) and energy expenditure (**d**) depending on the electric field strength and pulse duration.

**Figure 4 molecules-29-03428-f004:**
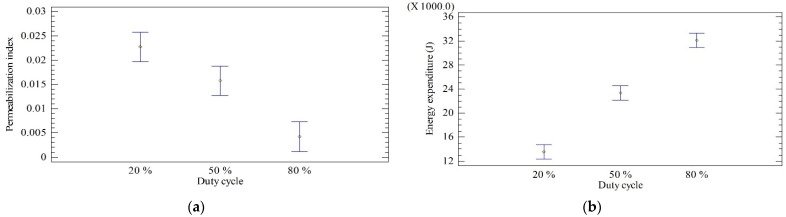
Averages according to duty cycle, Tukey HSD method with 95% confidence: (**a**) permeabilization index; (**b**) energy expenditure.

**Figure 5 molecules-29-03428-f005:**
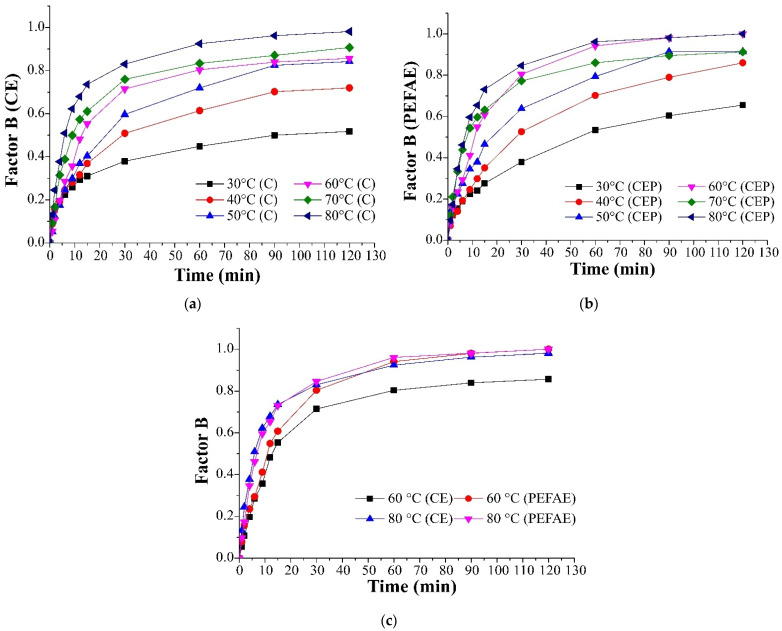
Factor B as a function of time at different temperatures: (**a**) conventional extraction; (**b**) PEF-assisted extraction; (**c**) comparison at 60 and 80 °C.

**Figure 6 molecules-29-03428-f006:**
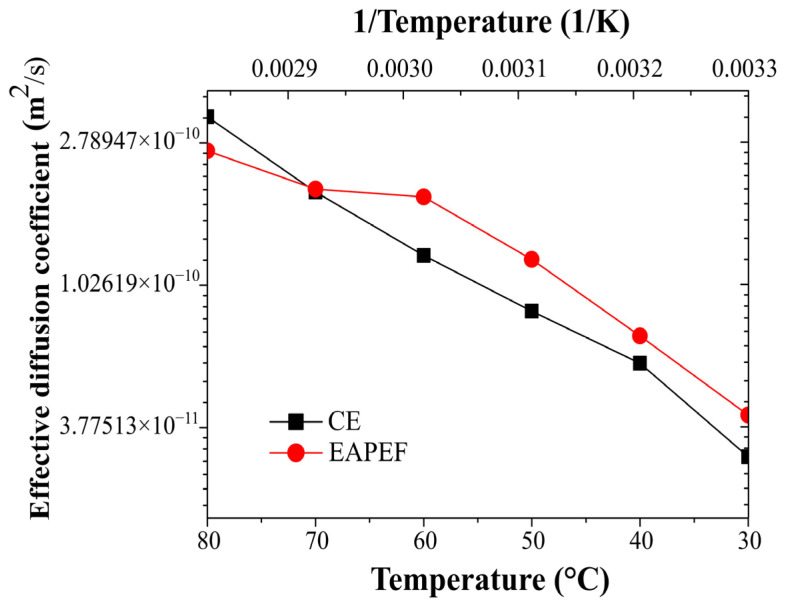
Effective solute diffusion coefficient as a function of temperature for conventional extraction and assisted extraction by pulsed electric field of fructans in water from cabuya.

**Figure 7 molecules-29-03428-f007:**
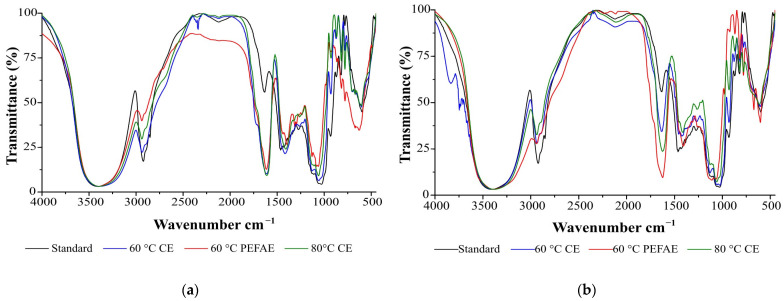
FTIR analysis of standard inulin and those obtained by PEFAE and CE: (**a**) inulin of high average polymerization degree $\left({In}_{H-APD}\right)$ and (**b**) low average polymerization degree (${In}_{L-APD})$.

**Figure 8 molecules-29-03428-f008:**
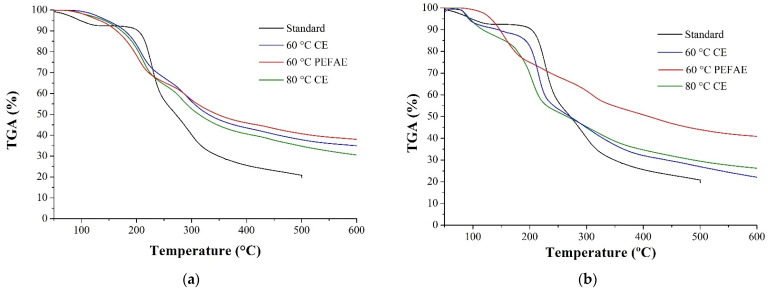
Thermogravimetric analysis (TGA) of standard inulin and the inulin obtained by PEFAE and CE: (**a**) inulin of high average polymerization degree $\left({In}_{H-APD}\right)$ and (**b**) low average polymerization degree (${In}_{L-APD})$.

**Figure 9 molecules-29-03428-f009:**
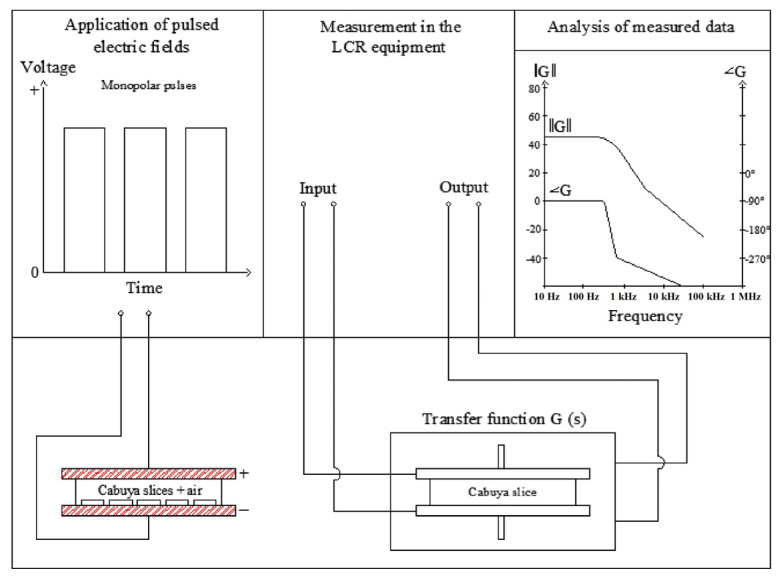
Scheme of the method used to determine the module and phase of the impedance.

**Table 1 molecules-29-03428-t001:** Comparison between PEF-assisted extraction and conventional extraction.

Method		PEFAE	CE	
Temperature (°C)		60	60	80
Effective solute diffusion coefficient (m^2^/s)		2.36 × 10^−10^	1.68 × 10^−10^	3.64 × 10^−10^
Free fructose concentration (g/L)		5.12	6.38	4.12
Inulin concentration (g/L)		43.86	38.41	42.64
Recovery (%)		12.49	93.33	32.63
Production (%)		1.42	9.49	3.62
${\mathrm{In}}_{H-\mathrm{APD}}$	Fraction at 20% *v*/*v* (%)	25.82	7.48	15.05
	Fraction at 40% *v*/*v* (%)	26.45	7.16	15.57
	Fraction at 60% *v*/*v* (%)	13.00	11.26	37.14
${\mathrm{In}}_{L-\mathrm{APD}}$	Fraction at 80% *v*/*v* (%)	34.72	74.10	32.24

**Table 2 molecules-29-03428-t002:** Comparison of the percentages of degradation of the inulin at 500 °C observed in Figure 8.

Inulin	Extraction Method	Degradation at 500 °C (%)
Standard	Standard	79.8
${In}_{H-APD}$	PEFAE at 60 °C	59.3
	CE at 60 °C	62.2
	CE at 80 °C	65.2
${In}_{L-APD}$	PEFAE at 60 °C	56.0
	CE at 60 °C	73.1
	CE at 80 °C	70.5

## Data Availability

The data presented in this study are available in article.

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
