# Peer review of "Pulsed Electric Field-Assisted Extraction of Inulin from Ecuadorian Cabuya (Agave americana)"

_molecules, 2024, doi:10.3390/molecules29143428_

Round 1

Reviewer 1 Report

Comments and Suggestions for Authors

The present study aimed to investigate the potential of the pulsed electric fields assisted extraction (PEFAE) to isolate inulin from cabuya (Agave americana). The effect of the electric field strength, number of pulses, pulse duration, duty cycle and extraction temperature were determined, and the characteristics of the extract and inulin obtained were compared to the one obtained by the conventional extraction. Generally, this is well designed study, the methods are appropriate for this type of a study, and some new and significant results were obtained (although PEF was not demonstrated to be superior to conventional extraction at all aspects). However, there are some concerns that need to be clarified.

Firstly, the abstract should be substantially modified – it needs to be summarized which parameters were varied, what was the purpose of that, etc. The conclusion should be added. Also, in my opinion, it should be stated first that the presence of inulin in extracts was confirmed by FTIR, and then the quality of isolated inulin should be mentioned. What do you mean (both in the abstract and in the main text) by “production of inulin”? At line 17, “project” should be replaced by “study”. At line 27, it should be “lower” instead of "lowest” since two of them were compared.

The conventional extraction is not clearly defined. Is it under all the same conditions excluding the use of PEF?

The part on Statistics is missing in “Methods” section and it should be added. Tukey HSD method, for example, is only mentioned in “Results”.

Line 39: “the seventh cause of death” – what does it refer to (colon cancer, obesity or diabetes)?

Line 300: What kind of spectra?

Lines 369-370: Equations should be added when the first time mentioned.

Line 471: Why 7100 RCF? The final concentration of hydrogen peroxide was 0.9% (30%v/v of a 3%v/v solution)?

Conclusions: I am not sure we can call the industrial production of inulin from cabuya at 60 °C a cold method.

Eventually, there are too many technical errors:

Title – ‘americana’ should be written with the first lowercase letter.

Percentage sign (%) should be written immediately after the number (without space).

In several places in the manuscript, decimal comma was used instead of a decimal point.

Figure numbers in the text are not correct in many places (lines 105, 167, 174 etc).

Line 189 – What does “5” stand for?

The exponents are not written correctly throughout the text (e.g., line 217 – it should be 10-2 instead of 10-2). Also, check how the exponents should be written in this journal (Ex or 10x).

In several figures it is written EAPEF instead of PEFAE.

Line 312 – the reference should be numbered.

Line 397 – it should be 33 instead of 33.

Comments on the Quality of English Language

There are no major issues detected. However, there are too many technical errors in the text that need to be corrected.

Author Response

  1. Firstly, the abstract should be substantially modified – it needs to be summarized which parameters were varied, what was the purpose of that, etc. The conclusion should be added. Also, in my opinion, it should be stated first that the presence of inulin in extracts was confirmed by FTIR, and then the quality of isolated inulin should be mentioned. What do you mean (both in the abstract and in the main text) by “production of inulin”? At line 17, “project” should be replaced by “study”. At line 27, it should be “lower” instead of "lowest” since two of them were compared. 

Abstract was modified, words at lines 17 and 27 were corrected 

  1. The conventional extraction is not clearly defined. Is it under all the same conditions excluding the use of PEF? 

Yes, the conventional extraction is under all the same conditions excluding the use of PEF. The CE method was included in the manuscript. 

  1. The part on Statistics is missing in “Methods” section and it should be added. Tukey HSD method, for example, is only mentioned in “Results”. 

The experimental designs employed in the research are included in section 3.2.2 and 3.2.3 of the section materials and methods. More details about the statistical analyses were included in the manuscript. 

  1. Line 39: “the seventh cause of death” – what does it refer to (colon cancer, obesity or diabetes)? 

It refers to diabetes, it was clarified on the text 

  1. Line 300: What kind of spectra? 

The infrared spectra of the isolated inulins powders from, it was included in the text 

  1. Lines 369-370: Equations should be added when the first time mentioned. 

Order of equations was changed as it is suggested 

  1. Line 471: Why 7100 RCF? The final concentration of hydrogen peroxide was 0.9% (30%v/v of a 3%v/v solution)? 

We expressed the units in rfc (relative centrifugal force) instead of rpm because rfc considers other variables than rpm and is more reproducible in research. Also, the 7100 rfc value allowed a correct precipitation of inulin. 

Yes, it was 0.9% v/v, it was changed in the text 

  1. Conclusions: I am not sure we can call the industrial production of inulin from cabuya at 60 °C a cold method. 

Conclusions were rewritten and improved with suggestions 

  1. Eventually, there are too many technical errors: 
  • Title – ‘americana’ should be written with the first lowercase letter. 

Updated 

  • Percentage sign (%) should be written immediately after the number (without space). 

Corrected in all the manuscript 

  • In several places in the manuscript, decimal comma was used instead of a decimal point. 

Corrected in all the manuscript 

  • Figure numbers in the text are not correct in many places (lines 105, 167, 174 etc). 

Updated 

  • Line 189 – What does “5” stand for? 

Eliminated 

  • The exponents are not written correctly throughout the text (e.g., line 217 – it should be 10-2 instead of 10-2). Also, check how the exponents should be written in this journal (Ex or 10x). 

Corrected in all the manuscript 

  • In several figures it is written EAPEF instead of PEFAE. 

Corrected in all the manuscript 

  • Line 312 – the reference should be numbered. 

Updated 

  • Line 397 – it should be 33 instead of 33. 

Updated 

Reviewer 2 Report

Comments and Suggestions for Authors

The article under appreciation is an interesting contribution and the study is well performed but some improvements could be made:

 - Introduction - authors are asked to highlight the originality of the work and show what they bring new with their experimental determinations that they have achieved.

- Discussions - the authors presented only their own results, without correlating them with the results from literature obtained by other researchers on the same subject.

Author Response

Response to Reviewer 2 Comments 

The article under appreciation is an interesting contribution and the study is well performed but some improvements could be made: 

 - Introduction - authors are asked to highlight the originality of the work and show what they bring new with their experimental determinations that they have achieved. 

Originality of the work was highlighted in the introduction 

- Discussions - the authors presented only their own results, without correlating them with the results from literature obtained by other researchers on the same subject. 

Other research was included in the discussions to compare the obtained results 

Reviewer 3 Report

Comments and Suggestions for Authors

The authors did a good work from an experimental point of view, and I recommend the article for publication after some major revisions.

More specific:

L16: Rewrite the abstract and give a clearer description.

L25: You can replace the 5000 V/cm with 5 kV/cm. Also, use commas for thousands.

L31: Use different keywords from the title.

L33: Reduce the paragraphs.

L101: Results and Discussion.

L144: Replace the commas with dots for decimals. Also, the same in the text and other figures. What does it mean the blue line in the a and c graphs?

L346: Section 3; sub-section 3.1.

L492: More details are needed for the different methods of analysis: equipment, model, company, description of analysis, and setting points.

L493: You have no data and results from differential scanning calorimetry (DSC) in the text.

L496: Rewrite the conclusions in a paragraph connecting the sentences. 

Comments on the Quality of English Language

Extensive editing of English language required.

Author Response

Response to Reviewer 3 Comments 

The authors did a good work from an experimental point of view, and I recommend the article for publication after some major revisions. 

More specific: 

L16: Rewrite the abstract and give a clearer description. 

The abstract was rewritten and improved 

L25: You can replace the 5000 V/cm with 5 kV/cm. Also, use commas for thousands. 

Recommendations were included in the manuscript 

L31: Use different keywords from the title. 

Recommendations were included in the manuscript 

L33: Reduce the paragraphs. 

Text was reduced 

L101: Results and Discussion. 

Subtitle was corrected 

L144: Replace the commas with dots for decimals. Also, the same in the text and other figures. What does it mean the blue line in the a and c graphs? 

Commas were replaced with dots in text and figures. The blue line is part of the graph that statistical software creates, the line divides the variables with or without significance. Ergo from the blue line to the right the variables are significant and those to the left are not significant. 

L346: Section 3; sub-section 3.1. 

Updated 

L492: More details are needed for the different methods of analysis: equipment, model, company, description of analysis, and setting points. 

Details of the different methods and equipments were added to the manuscript. 

L493: You have no data and results from differential scanning calorimetry (DSC) in the text. 

The text was removed. 

L496: Rewrite the conclusions in a paragraph connecting the sentences.  

Conclusions were rewritten in a single paragraph 

Round 2

Reviewer 1 Report

Comments and Suggestions for Authors

The authors have significantly modified and improved the initial version of their manuscript. I support this manuscript to be published in the current form.

Comments on the Quality of English Language

No major issues were detected.

Reviewer 3 Report

Comments and Suggestions for Authors

The paper has been revised according to the suggestions and criticisms of the reviewers. In this revised version, the paper has improved its quality and I recommend the article for publication.